# Influence of Oral Administration of Lactic Acid Bacteria Metabolites on Skin Barrier Function and Water Content in a Murine Model of Atopic Dermatitis

**DOI:** 10.3390/nu10121858

**Published:** 2018-12-01

**Authors:** Yoshihiro Tokudome

**Affiliations:** Laboratory of Dermatological Physiology, Department of Pharmaceutical Sciences, Faculty of Pharmacy and Pharmaceutical Sciences, Josai University, 1-1 Keyakidai, Sakado, Saitama 350-0295, Japan; tokudome@josai.ac.jp; Tel.: +81-49-271-8140

**Keywords:** oral administration, lactic acid bacteria metabolites, skin barrier function, water content, atopic dermatitis

## Abstract

The effects of orally administered lactic acid bacteria metabolites on skin were studied using an atopic dermatitis-like murine model generated by feeding HR-AD to mice. Lactic acid bacteria metabolites were obtained by inoculating and culturing soy milk with 35 strains of 16 species of lactic acid bacteria. The atopic dermatitis-like murine model was generated by feeding HR-AD to HR-1 mice for 40 days. The skin condition of HR-AD-fed mice worsened compared with normal mice, showing reduced water content in the stratum corneum, increased transepidermal water loss (TEWL), reduced ceramide AP content in the stratum corneum, and increased epidermis thickness. When HR-AD-fed mice were orally administered a raw liquid containing lactic acid bacteria metabolites, water content in the stratum corneum, TEWL, ceramide AP content in the stratum corneum, and epidermis thickness improved. To determine the active components responsible for these effects, filtrate, residue, and lipid components extracted from the raw liquid containing lactic acid bacteria metabolites were examined. While water-soluble components and residue obtained after filtration had no effects, the lipid fraction showed similar effects to the raw liquid. These findings suggest that lactic acid bacteria metabolites improve skin injury in an atopic dermatitis-like murine model.

## 1. Introduction

One hundred trillion enteric bacteria inhabit the human intestines, among which lactic acid bacteria are one example. However, orally ingested lactic acid bacteria may have difficulty proliferating or colonizing the intestines because they differ from the indigenous intestinal bacteria (intestinal flora). Because ingested lactic acid bacteria are quickly excreted from the body, they have limited time to exert their effects. However, lactic acid bacteria metabolize sugar to produce lactic acid metabolites, which form as fermentation products. Reports on the effectiveness of lactic acid metabolites as biogenics have increased in recent years [1]. These biogenics activate immune function in the intestines, an effect that is not mediated by intestinal flora, and decrease reactive oxygen species levels [2]. These lactic acid bacteria metabolites are present in foods consumed in our everyday diet, such as cheese, yogurt, and pickles. Increasing interest has led to lactic acid bacteria metabolites becoming a major subject of research. Mitsuoka et al. [3,4,5,6] demonstrated the importance of food components that function directly or via intestinal flora in biological regulation, host defense, disease prevention, recovery, and aging that act through immunostimulation, cholesterol-lowering effects, blood pressure-lowering effects, regulation of intestinal function, antitumor effects, antithrombotic effects, and hematopoiesis.

The incidence of atopic dermatitis is increasing. Dry skin, skin inflammation, and pruritus are the main symptoms of atopic dermatitis [7], and are often accompanied by the overproduction of Immunoglobulin E (IgE) [8]. The pathogenic mechanisms of atopic dermatitis are complex. Reports indicate that there is less ceramide in the stratum corneum of atopic dermatitis patients compared to normal subjects [9,10,11]. Filaggrin gene mutations have also been implicated as a cause of atopic dermatitis [12,13]. However, the fundamental cause and treatments have yet to be identified and remain a major topic of research worldwide. A number of studies have reported atopic dermatitis models using NC/Nga mice [14] and HR-1 hairless mice fed a HR-AD diet [15,16]. NC mice with dermatitis exhibit decreased water content in the stratum corneum, acanthosis, and changes in the content and composition of ceramide in the skin [14]. Similarly, HR-1 hairless mice that were fed HR-AD exhibited decreased water content in the stratum corneum, increased scratching, acanthosis, and increased total IgE levels in the blood [17]. HR-AD differs from a normal diet in that there is less magnesium [15].

Recent studies indicate that there is an association between the intestinal environment and skin condition [18,19]. As lactic acid bacteria metabolites are thought to exert various effects via their regulation of the intestinal environment, this study evaluated the effectiveness of orally administered lactic acid bacteria metabolites in mice with an atopic dermatitis-like condition induced by HR-AD by comparing the water content in the stratum corneum, transepidermal water loss (TEWL), and epidermis thickness with those of normal mice.

## 2. Materials and Methods

### 2.1. Metabolites

Lactic acid bacteria metabolites were provided by KOEI Science Laboratory Co. Ltd. (Wako City, Saitama, Japan). Soy milk was heat-treated at 95 °C for 1 h, cooled, and inoculated with 35 strains of 16 species of lactic acid bacteria and cultured at 37 °C for 120 h in a static culture (Figure 1). The culture was subsequently sterilized at 95 °C for 30 min. The resulting lactic acid bacteria-fermented liquid was referred to as the lactic fermentation product (LFP). The LFP contained soy milk substrate and killed lactic acid bacteria. The LFP was filtered to obtain lactic fermentation secretion (LFS) and the residue of LFP (R-LFP). LFS is characterized by the absence of soy milk substrate and killed lactic acid bacteria. The R-LFP was heated to obtain the solid matter. Chloroform and methanol were added to the solid matter to recover lipids, and this was referred to as the lipid components of LFP (LC-LFP). When the weight of the LFP was set to 100%, the weight of the LFS was 80%, R-LFP was 18%, and LC-LFP was 2%. Each sample was adjusted to the same volume with water.

### 2.2. Animals and Rearing

Male 6-week-old hairless mice were purchased from Hoshino Laboratory Animals, Inc. (Bando, Ibaraki, Japan) and acclimated for 1 week. Mice assigned to the test group were provided access to HR-AD (NOSAN Corporation, Yokohama, Kanagawa, Japan) ad libitum, while those in the normal group were provided access to Labo MR Stock (NOSAN Corporation, Yokohama, Kanagawa, Japan) ad libitum. All mice were also provided access to tap water ad libitum. The mice were reared in a room maintained on a 12-h light/dark cycle (7 a.m. to 7 p.m.) at 20–26 °C. All animal experiments and maintenance were approved by the Animal Research Committee of Josai University (approval date: 3 April 2018, approval number: JU18094).

### 2.3. Grouping of Animals and Administration Methods

The atopic dermatitis-like murine model was generated by continuously feeding HR-1 hairless mice with HR-AD for approximately 40 days [15]. Mice were divided into the following 5 groups with 6 mice per group according to their weight on the day of the experiment and treatment to be administered: normal group (no treatment), control group, LFP group (treatment with LFP), LFS group (treatment with the LFP filtrate), R-LFP group (treatment with the residue of LFP), and LC-LFP group (treatment with lipid components of LFP) (Table 1). Each treatment was orally administered through a tube at a volume of 0.3 mL per animal for 28 days. Body weight, water content in the stratum corneum, and TEWL were measured. On day 28, animals were sacrificed and their skin was frozen, sectioned, and stained with hematoxylin and eosin (HE) for light microscopic observation of acanthosis. The amount of LFS or LC-LFP orally administered to each group was based on the amount of LFS or LC-LFP obtained from LFP. That is, when 0.3 mL of LFP was administered per animal, the amount of LFS or LC-LFP obtained from 0.3 mL of LFP was administered. Ethanol at 10.8%, which was used as a vehicle for these fermented substances, was administered orally to the control group, while distilled water was administered to the normal group.

### 2.4. Measurement of Transepidermal Water Loss and Water Content of the Stratum Corneum

All measurements were performed in triplicate for each mouse, and the mean values were calculated. Water content of the stratum corneum and skin was measured using a Cutometer MPA580 (Courage & Khazaka, Cologne, Germany), and TEWL was assessed using a VAPO SCAN AS-VT100 RS (Asahi Techno Lab. Ltd., Yokohama, Kanagawa, Japan).

### 2.5. Hematoxylin and Eosin Staining

Cryosections were prepared from tissue samples embedded in 10% formalin in phosphate buffered saline (PBS). Skin sections (10 μm) were stained with hematoxylin and eosin (HE) and analyzed for structural differences using light microscopy.

### 2.6. Collection of the Stratum Corneum

A glass slide with one drop of cyanoacrylate adhesive was pressed onto the back of a mouse under anesthesia for 1 min and then peeled off slowly to obtain a sample of the stratum corneum.

### 2.7. Measurement of Stratum Corneum Mass

The glass slide used to collect the stratum corneum was soaked in *N,N*-dimethylformamide, removed, and the resulting solution was sonicated for 15 min. The sonicated solution was passed through a filter that was weighed beforehand. The filter was dried for one week under vacuum. After confirming that the weight of the filter was no longer changing, the weight of the stratum corneum was calculated by subtracting the original weight of the filter from that after filtration.

### 2.8. Extraction of Lipids and Quantification of Ceramide (AP)

The glass slide used to collect the stratum corneum was soaked in hexane and ethanol (95:5), removed, and the resulting solution was sonicated at 37 °C for 20 min. After filtering, the solution was dried under nitrogen gas. Chloroform–methanol (2:1, *v*/*v*) was added to the residue to extract lipids. The extracted sample was separated on a high performance thin layer chromatography (HPTLC) plate (Silica Gel 60, Merck, Darmstadt, Germany). The plate was developed twice using chloroform–methanol–acetic acid (190:9:1, *v*/*v*) as an eluent. Subsequently, 10% CuSO_4_ and 8% H_3_PO_4_ aqueous solution were sprayed onto the plate, and the plate was heated at 180 °C for 10 min in a plate heater for visualization. Ceramide was quantified based on the density of the visualized band.

### 2.9. Measurement of Epidermal Thickness

The thickness of the epidermis was measured at the horizontal midpoint of each visual field. Approximately 50 individual measurements were made along the wound margin for each histological section, and the mean thickness was calculated.

### 2.10. Data and Statistical Analysis

All results are expressed as mean ± SD of four or five experiments. Statistical analysis was performed using Tukey’s multiple comparisons test (JMP ver. 13, SAS Institute Inc., Cary, NC, USA).

## 3. Results

Figure 2 shows the changes in body weight of HR-1 hairless mice that were fed HR-AD or a regular diet for 40 days. On the final day, the average weight in the normal group was 31.8 g, and that in the HR-AD-fed group was 27.1 g. Forty days after the start of rearing, various preparations of lactic acid bacteria metabolites were orally administered, and body weight was measured over the 28-day administration period (Figure 2). On the final day, the average weight in the normal group, which was fed a regular diet, was 33.0 g, and that among HR-AD-fed mice was 26.5 g, 25.7 g, 25.5 g, 26.0 g, and 26.4 g for the control, LFP, LFS, R-LFP, and LC-LFP administration groups, respectively. No significant differences were observed among the groups.

Figure 3 shows the change in water content of the stratum corneum of hairless mice that were fed HR-AD and subsequently administered lactic acid bacteria metabolites. The average water content in the stratum corneum (arbitrary units, AU) immediately before administration of lactic acid bacteria metabolites was 56.6 AU in the normal group and 27.0 AU in the HR-AD-fed groups, and this difference was significant. This suggests that mice that were fed HR-AD developed an atopic dermatitis-like condition. Among the HR-AD-fed groups that received lactic acid bacteria metabolites, water content in the stratum corneum started to increase from 10 days after the initiation of oral administration in the LFP and LC-LFP groups, and reached 43.5 and 37.8 AU 28 days after the initiation of administration, respectively. The water content in the stratum corneum was 24.2 AU in the control group and 26.6 AU in the R-LFP group, both of which were significantly lower than the values in the lactic acid bacteria metabolite groups. In the LC-LFP group, water content in the stratum corneum tended to be higher than that in the other groups that received lactic acid bacteria metabolites until 14 days after the initiation of administration, but the difference was not significant.

Figure 4 shows the effect of lactic acid bacteria metabolites on TEWL in mice that were fed HR-AD. In the normal group, which was fed a regular diet, TEWL (g/m^2^/h) was 5.5 g/m^2^/h at the start of the study and did not vary greatly during the study period. In contrast, TEWL immediately increased after mice were fed HR-AD, reaching 24.1 g/m^2^/h after 40 days on a HR-AD diet, which was a significant increase. The elevated TEWL was reduced after oral administration of LFP and LC-LFP to 10.4 and 19.9 g/m^2^/h, respectively, which was a significant reduction compared with that in the control group. In contrast, TEWL in the LFS and R-LFP groups was 18.9 and 22.9 g/m^2^/h after oral administration, respectively, and the reduction was not significantly different from that in the control group.

Figure 5 shows the ceramide AP content in the stratum corneum at the completion of the study. Mice that were fed HR-AD had a ceramide AP content of 0.57 mmol/g in the stratum corneum. In the LFP, LC-LFP, and LFS groups, the ceramide AP content in the stratum corneum was 2.25, 2.08, and 1.90 mmol/g stratum corneum, respectively, and was significantly higher than that in the control group. These values were similar to that in the normal group, which was fed a regular diet (2.29 mmol/g stratum corneum). In contrast, the ceramide AP content in the stratum corneum of the R-LFP group was 0.78 mmol/g stratum corneum.

Figure 6a–f shows images of HE-stained epidermal sections from mice that were fed HR-AD prepared 30 days after the start of administration of lactic acid bacteria metabolites. Compared to the epidermis of the normal group, the control group, which was fed HR-AD and orally administered vehicle, had a thicker epidermis. The LFP and LC-LFP groups had reduced acanthosis compared with the control group and similar epidermis thickness to that in the normal group. In contrast, the LFS and R-LFP groups demonstrated similar levels of acanthosis to that observed in the control group. Quantification of the thickness of the epidermis across the examined groups is shown in Figure 6g. As observed in the images, oral administration of LFP and LC-LFP significantly suppressed acanthosis compared with the control group, while the LFS and R-LFP groups demonstrated similar levels of acanthosis to that observed in the control group.

## 4. Discussion

In this study, mice were fed HR-AD to generate an atopic dermatitis model to study the effects of oral administration of various preparations of lactic acid bacteria metabolites on the skin. First, the effect of lactic acid bacteria metabolites on body weight was examined. Mice that were fed HR-AD, the atopic dermatitis model, lost weight compared with mice that were fed a regular diet.

Studies have reported that continuous feeding and rearing of hairless mice with HR-AD leads to atopic dermatitis (AD)-like symptoms, which arise mainly due to a lack of polyunsaturated fatty acids (*n-6* PUFAs) but not due to a lack of magnesium [15,16]. A more recent study reported that a lack of polyunsaturated fatty acids and starch causes atopic dermatitis-like symptoms [20].

Mice reared with this diet with atopic dermatitis-like symptoms were subsequently used to evaluate the effectiveness of lactic acid bacteria metabolites for reducing these symptoms. Water content in the stratum corneum and TEWL were improved in the LFP and LC-LFP groups (Figure 3 and Figure 4), but were similar in the LFS and R-LFP groups compared with the normal group. Ceramide levels in the stratum corneum tend to be reduced in atopic dermatitis patients [9,11]. Ceramide, along with cholesterol and fatty acids, is present intercellularly in the stratum corneum. A number of studies have demonstrated that among the lipids in the stratum corneum, ceramide is one of the most important for barrier function [21,22]. Water is also present in intercellular lipids in the stratum corneum [23,24,25]. The results of the present study suggest that a reduction in lipids, such as ceramide, in the skin, in particular in the stratum corneum, of those with atopic dermatitis may weaken the barrier function, resulting in increased TEWL and decreased water content in the stratum corneum. Therefore, this study examined the ceramide AP content in each group. Ceramide AP content in the control group, which was fed HR-AD but did not receive lactic acid bacteria metabolites, decreased to approximately one-quarter of that in the normal group (Figure 5). Ceramide AP content in the stratum corneum tended to be high in mice with TEWL and water content in the stratum corneum, and these values were almost correlated. A previous study reported that when ceramide EOS is concurrently present, ceramide AP in the stratum corneum contributes to the stabilization of the lamellar structure of the stratum corneum [26]. Therefore, lactic acid bacteria metabolites may have affected the barrier function of the stratum corneum by changing the ceramide AP content in the stratum corneum. A more detailed study is needed to investigate this hypothesis. Interestingly, levels of ceramides other than ceramide AP, such as NS, NP, and AS, and cholesterol and fatty acids did not vary greatly following administration of lactic acid bacteria metabolites (data not shown). The reason for this is unclear and should be examine in future studies. The water content in the stratum corneum is greatly influenced by natural moisturizing factors and keratin in the stratum corneum [27]. It is possible that lactic acid bacteria metabolites may affect epidermal differentiation and the production of natural moisturizing factors in the stratum corneum, including amino acids. Future studies should quantify these factors to confirm this hypothesis.

The thickness of the epidermis is reportedly increased in mice with atopic dermatitis-like conditions [28]. The epidermis was thickened in mice that were fed HR-AD in this study. Acanthosis is typically caused by the abnormal growth of epidermal cells and a skin inflammatory response. While lactic acid bacteria metabolites have been shown to suppress these processes, the detailed mechanisms of the suppression are not known. To identify the substances in lactic acid bacteria metabolites that may be responsible for its suppression of atopic dermatitis-like symptoms, several fractions of the metabolites were examined. Administration of the water-soluble fraction, which was obtained by filtering lactic acid bacteria metabolites, did not improve TEWL, water content in the stratum corneum, or acanthosis, suggesting that compounds that improve the skin condition of animals with an atopic dermatitis-like condition are not contained in the water-soluble fraction. In contrast, administration of the residual components, in particular the lipid fraction, resulted in effects that were similar to that observed using the raw liquid containing lactic acid bacteria metabolites. Therefore, the components responsible for improving atopic dermatitis-like symptoms are likely compounds that were extracted with chloroform or methanol. As a side note, the current study also demonstrated that the residual components were not potent. Future studies should identify the effector components by determining their structures. Lactic acid bacteria metabolites may contain unknown compounds that are produced by fermenting soy milk with lactic acid bacteria that are not contained in soy milk or lactic acid bacteria themselves. Future studies are needed to explore these possibilities. Currently, treatment of atopic dermatitis is achieved using steroids, tacrolimus, antihistamines, antiallergic drugs, and external moisturizing agents. Daily life guidance is also reportedly useful. In this study, oral administration of lactic acid bacteria metabolites improved the clinical symptoms of atopic dermatitis, such as dry skin and epidermal hyperplasia. That atopic dermatitis-like conditions were improved by oral administration of lactic acid bacteria metabolites may provide important insight for the future development of treatments for atopic dermatitis.

## Figures and Tables

**Figure 1 nutrients-10-01858-f001:**
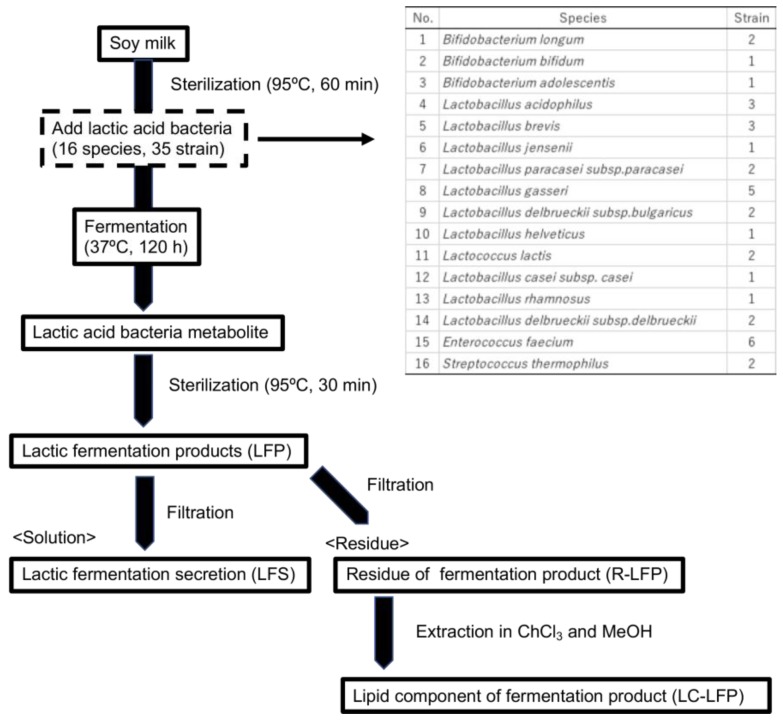
Method for the production of lactic acid bacteria metabolites and the species of microorganisms used.

**Figure 2 nutrients-10-01858-f002:**
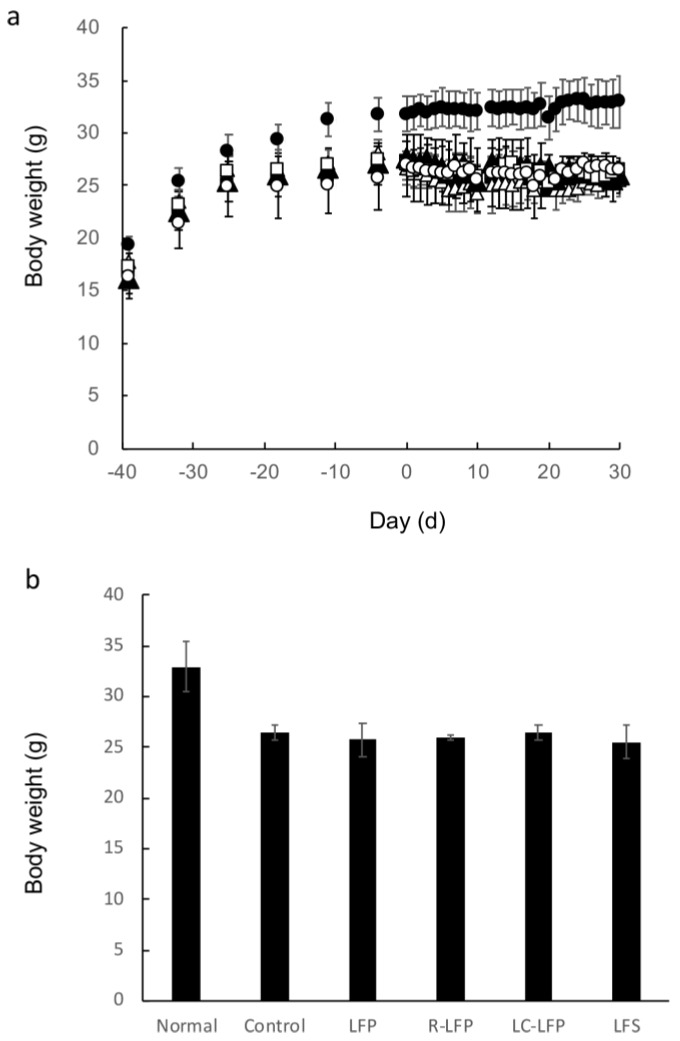
Changes in body weight in normal mice and HR-AD-fed mice after oral administration of lactic acid bacteria metabolites over time (**a**), and after 40 days (**b**). Data represent mean and standard deviation (*n* = 4 to 6). Symbols: closed circle, normal (regular diet fed, distilled water); open circle, control (HR-AD fed, 10.8% ethanol in water); closed square, lactic fermentation products (HR-AD fed, LFP); closed triangle, residue of LFP (HR-AD fed, LFP); open square, lipid components of LFP (HR-AD fed, LC-LFP); open triangle, lactic fermentation secretion (HR-AD fed, LFS).

**Figure 3 nutrients-10-01858-f003:**
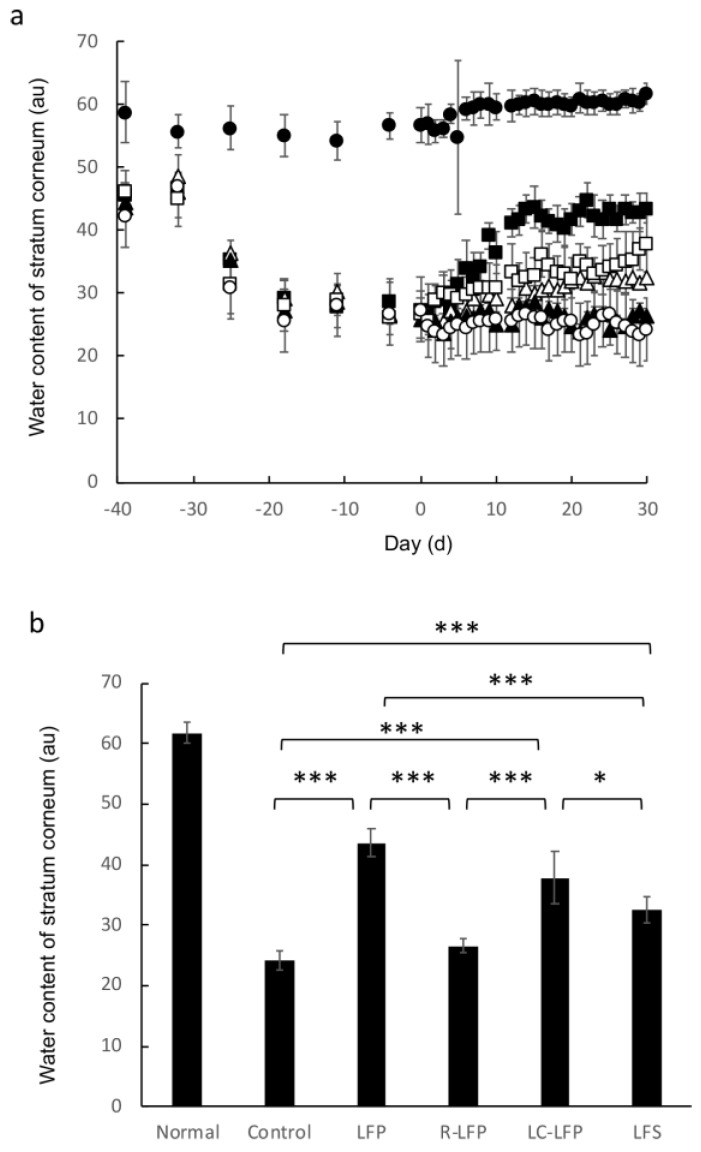
Water content of the stratum corneum of regular and HR-AD-fed mice after oral administration of lactic acid bacteria metabolites over time (**a**), and after 40 days (**b**). Data represent mean and standard deviation (*n* = 4 to 6). * *p* < 0.05, *** *p* < 0.001 (Tukey’s post-hoc test). Symbols: closed circle, normal (regular diet fed, distilled water); open circle, control (HR-AD fed, 10.8% ethanol in water); closed square, lactic fermentation products (HR-AD fed, LFP); closed triangle, residue of LFP (HR-AD fed, LFP); open square, lipid components of LFP (HR-AD fed, LC-LFP); open triangle, lactic fermentation secretion (HR-AD fed, LFS).

**Figure 4 nutrients-10-01858-f004:**
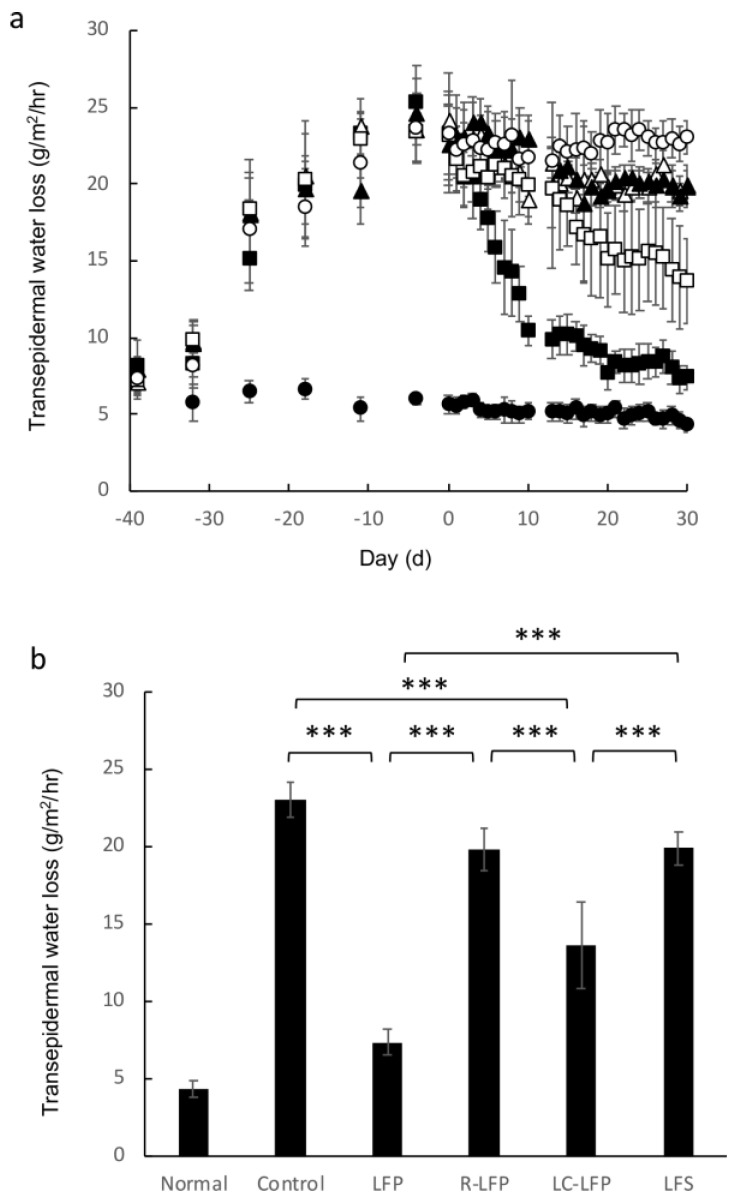
Transepidermal water loss in regular and HR-AD-fed mice after oral administration of lactic acid bacteria metabolites over time (**a**), and after 40 days (**b**). Data represent mean and standard deviation (*n* = 4 to 6). * *p* < 0.05, *** *p* < 0.001 (Tukey’s post-hoc test) . Symbols: closed circle, normal (regular diet fed, distilled water); open circle, control (HR-AD fed, 10.8% ethanol in water); closed square, lactic fermentation products (HR-AD fed, LFP); closed triangle, residue of LFP (HR-AD fed, LFP); open square, lipid components of LFP (HR-AD fed, LC-LFP); open triangle, lactic fermentation secretion (HR-AD fed, LFS).

**Figure 5 nutrients-10-01858-f005:**
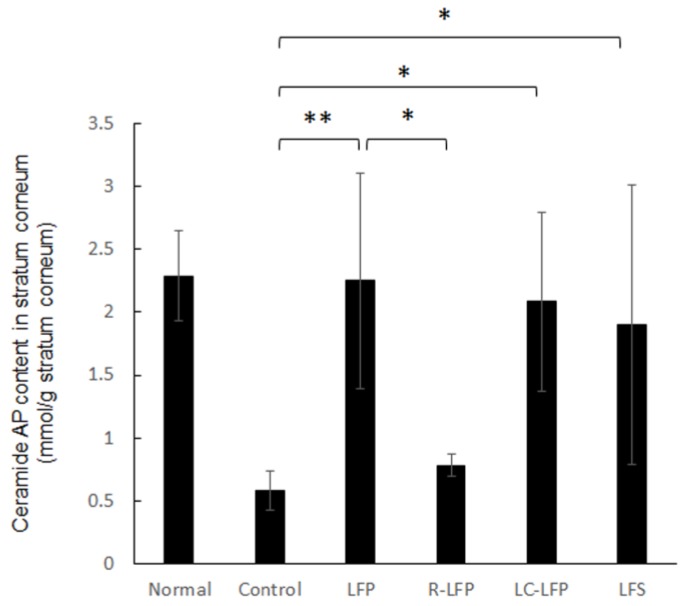
Ceramide AP content in the stratum corneum of regular and HR-AD-fed mice after oral administration of lactic acid bacteria metabolites for 40 days. * *p* < 0.05, ** *p* < 0.01 (Tukey’s post-hoc test).

**Figure 6 nutrients-10-01858-f006:**
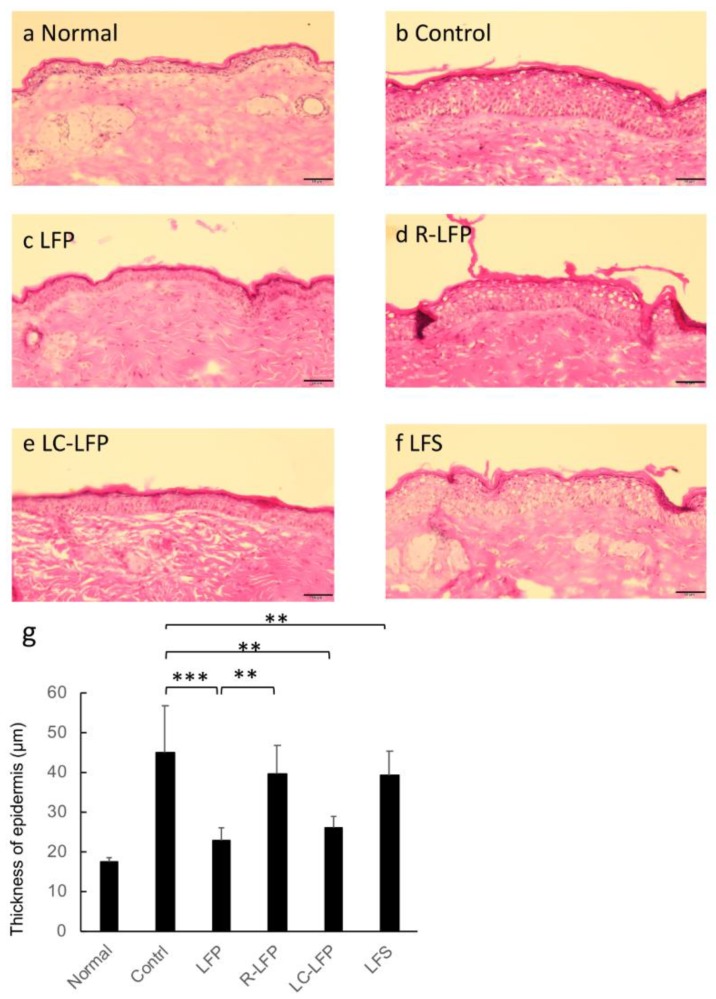
(**a**–**g**) Histological images of skin sections from regular and HR-AD-fed mice after oral administration of lactic acid bacteria metabolites for 40 days. Skin samples were fixed with 10% formalin in PBS, sectioned, and stained with hematoxylin and eosin. (**g**) Quantification of the thickness of the epidermis after 40 days. ** *p* < 0.01, *** *p* < 0.001 (Tukey’s post-hoc test). Groups: normal (regular diet fed, distilled water), control (HR-AD fed, 10.8% ethanol in water), lactic fermentation products (HR-AD fed, LFP), residue of LFP (HR-AD fed, LFP), lipid components of LFP (HR-AD fed, LC-LFP) and lactic fermentation secretion (HR-AD fed, LFS) groups. Scale bar represents 50 μm.

**Table 1 nutrients-10-01858-t001:** Experimental groups in this study.

Experimental Group	Oral Administration (0.3 mL/day)	Diet
Normal	Distilled water	LaboMR Stock (Regular diet)
Control	10.8% ethanol in water	HR-AD
Lactic fermentation product (LFP)	LFP/10.8% ethanol in water	HR-AD
Residue of lactic fermentation product (R-LFP)	R-LFP/10.8% ethanol in water	HR-AD
Lipid component of lactic fermentation product (LC-LFP)	LC-LFP/10.8% ethanol in water	HR-AD
Lactic fermentation secretion (LFS)	LFS/10.8% ethanol in water	HR-AD

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
