# Peer review of "Influence of Oral Administration of Lactic Acid Bacteria Metabolites on Skin Barrier Function and Water Content in a Murine Model of Atopic Dermatitis"

_nutrients, 2018, doi:10.3390/nu10121858_

Reviewer 1 Report

Interesting study which is worth to be published.

As a dermatologist who works both scientifically and clinically I would like the authors to better identify the therapeutic implication. 

Author Response

Thank you for your comments.

Reviewer 2 Report

This paper is of great interest, but I think there are a number of changes that would make it more valuable to the reader.

   I believe that the paper should be edited for style.   It is so difficult to write English when it is not one's first language but I find the phrasing very hard to follow.  The paper is difficult to read and therefore misses having the impact that it deserves.  I recommend consultation with an English grammarian and editor to help make the sentence structures more clear.

Clearly the authors are very familiar with the HR-1 mice and the ability of the HR-AD diet to induce a pruritic dermatitis very similar to atopic dermatitis.   But most of the readers may be less than familiar with this mouse model.  I think it is important to briefly describe the model, the HR-AD diet and how it differs from a normal diet (in general terms), and how the diet induces the disease.  Then the experiments become much more meaningful.  There is a minor comment about this in the discussion (low polyunsaturated fatty acids) but it is not sufficient.  this information needs to be in more detail in the introduction.

Figure 1:  the text is much too small.  this figure needs to be enlarged so that the text is readable at a normal magnification of the paper.

Figure 2:  this figure also needs to be enlarged and the figure legend clarified.  There is no description of what figure a is or what.figure b is.   I can guess, but the figure legend needs to be crystal clear.

Figure 3:  this figure, at least graph a, needs to be enlarged.

Figure 4:  as for figure 3,graph a needs to be enlarged

The authors need to explain why they chose ceramide AP and furthermore, for the readership, they need to define it.   why not the other ceramides that they mentioned in the discussion section?   The readers need to know why measuring ceramide AP is important for these experiments.  They alluded to some other species of ceramide briefly in the discussion as a data not shown. Frankly, I would like to see the data.  As a reviewer, I don't like to see the phrase "data not shown"

Figure 5:  this figure is so important, yet the presentation is quite erratic.  some of the images have imbedded information and some do not.  Furthermore, the magnification among the various images seems to differ.   I would either imbed the information in all the images, or none.  And these images could be improved with regard to quality.  They are very dark.  Why not brighten them up?  And give the magnification.

I think we need more information about the weight loss in these mice, and why the weight loss occurred.   was the feed intake down because of intense itch?  Was the caloric content of the HR-AD diet much lower than the normal diet?  Are the changes in the skin just related to dietary deficiency?  I think we need a more robust discussion about why this is a good model for atopic dermatitis.   This model has been published before, but readers need to understand the relevance without having to pull all the references. 

I would also like to see in the discussion more rationalization about why this is a good model for the natural expression of atopic dermatitis.   What is very clear in the natural disease is that there are two major factors:  a dysregulated immune system and a barrier defect.  I would like to see a little more discussion about relevance of this model to the natural disease.  and any comments as to the immunological findings in this model.  There is very brief mention about IgE but IgE is such a small part of the pathogenesis of atopic dermatitis

Last but not least, we could use more discussion, or at least more clarification of the role of the .

In summary, there is much of interest in this paper. I think with some editing and some clarification, this paper will be a great contribution.to the literature.

Author Response

This paper is of great interest, but I think there are a number of changes that would make it more valuable to the reader.

Dear Editor and Reviewer

Thank you for your great comments. I will reply one by one about the comment. Thank you.

1.    I believe that the paper should be edited for style.   It is so difficult to write English when it is not one's first language but I find the phrasing very hard to follow.  The paper is difficult to read and therefore misses having the impact that it deserves.  I recommend consultation with an English grammarian and editor to help make the sentence structures more clear.

Response: Thank you for your suggestion. The paper has been revised and reformatted and edited by a Native English editor.

2.    Clearly the authors are very familiar with the HR-1 mice and the ability of the HR-AD diet to induce a pruritic dermatitis very similar to atopic dermatitis.   But most of the readers may be less than familiar with this mouse model.  I think it is important to briefly describe the model, the HR-AD diet and how it differs from a normal diet (in general terms), and how the diet induces the disease.  Then the experiments become much more meaningful.  There is a minor comment about this in the discussion (low polyunsaturated fatty acids) but it is not sufficient.  this information needs to be in more detail in the introduction.

Response: Thank you for your suggestion. I have added a description of the HR-1 mouse model of atopic dermatitis to the Introduction section. I added the known thing of HR - AD and indicated it with underline, yellow line. 

3.    Figure 1:  the text is much too small.  this figure needs to be enlarged so that the text is readable at a normal magnification of the paper.

Response: Thank you for your suggestion. I have enlarged Figure 1 in the revised version of the manuscript.

4.    Figure 2:  this figure also needs to be enlarged and the figure legend clarified.  There is no description of what figure a is or what.figure b is.   I can guess, but the figure legend needs to be crystal clear.

Response: Thank you for your suggestion. I have enlarged Figure 2 and revised the description in the figure legend.

5.    Figure 3:  this figure, at least graph a, needs to be enlarged.

Response: Thank you for your suggestion. I have enlarged Figure 3 in the revised version of the manuscript.

6.    Figure 4:  as for figure 3,graph a needs to be enlarged

Response: Thank you for your suggestion. I have enlarged Figure 4 in the revised version of the manuscript.

7.    The authors need to explain why they chose ceramide AP and furthermore, for the readership, they need to define it.   why not the other ceramides that they mentioned in the discussion section?   The readers need to know why measuring ceramide AP is important for these experiments.  They alluded to some other species of ceramide briefly in the discussion as a data not shown. Frankly, I would like to see the data.  As a reviewer, I don't like to see the phrase "data not shown"

Response: Thank you for your suggestion. Ceramide AP has been reported to be involved in the stabilization of the barrier function. I indicated this with a yellow line and underline (Line #269 to 270). Actually, I have quantified Ceramide NS, AS, AP. However, because I can not explain it, I decided not to show data. It is shown here. We will rewrite if it should be rewrite in the manuscript.

Fig. Ceramide content in stratum corneum

8.    Figure 5:  this figure is so important, yet the presentation is quite erratic.  some of the images have imbedded information and some do not.  Furthermore, the magnification among the various images seems to differ.   I would either imbed the information in all the images, or none.  And these images could be improved with regard to quality.  They are very dark.  Why not brighten them up?  And give the magnification.

Response: Thank you for your comment and suggestions. I assume the reviewer means Figure 6 rather than Figure 5 based on his/her description of the contents of the figure. I have revised Figure 6 so that all labels are embedded in the images. All images are provided at the same magnification, as can be seen from the scale bars in each image. I have provided the length of the scale bar rather than the magnification because I think that this is more universal. I have also brightened the images.

9.    I think we need more information about the weight loss in these mice, and why the weight loss occurred.   was the feed intake down because of intense itch?  Was the caloric content of the HR-AD diet much lower than the normal diet?  Are the changes in the skin just related to dietary deficiency?  I think we need a more robust discussion about why this is a good model for atopic dermatitis.   This model has been published before, but readers need to understand the relevance without having to pull all the references. 

Response: Thank you for your suggestions. I have added an explanation for the weight loss observed in these mice and the effect on their skin to the Discussion section. I have also added an explanation about why this is a good model for atopic dermatitis to the Introduction section.  Add sentence is underlined.About weight loss, details are unknown. Some nutrients may be involved.I do not measure the feeding amount, but I think that it may be the same.This model has been published as several papers. Reference paper number is 15, 16, 17 and 20,

10.  I would also like to see in the discussion more rationalization about why this is a good model for the natural expression of atopic dermatitis.   What is very clear in the natural disease is that there are two major factors:  a dysregulated immune system and a barrier defect.  I would like to see a little more discussion about relevance of this model to the natural disease.  and any comments as to the immunological findings in this model.  There is very brief mention about IgE but IgE is such a small part of the pathogenesis of atopic dermatitis

Response: Thank you for your suggestion. There is no report on the influence of animals fed HR-Ad on the immune system. Therefore, we could not discuss the immune system etc. in this paper. I think that it is a future task.

11.  Last but not least, we could use more discussion, or at least more clarification of the role of the .

Response: I was unable to respond to this comment because it is not clear from the comment what the reviewer was requesting a clarification for.

       12. In summary, there is much of interest in this paper. I think with some editing and some clarification, this paper will be a great contribution.to the literature.

  Response: Thank you for comments to improve my manuscript.